# Exposure of Human Skin Organoids to Low Genotoxic Stress Can Promote Epithelial-to-Mesenchymal Transition in Regenerating Keratinocyte Precursor Cells

**DOI:** 10.3390/cells9081912

**Published:** 2020-08-18

**Authors:** Sophie Cavallero, Renata Neves Granito, Daniel Stockholm, Peggy Azzolin, Michèle T. Martin, Nicolas O. Fortunel

**Affiliations:** 1Laboratoire de Génomique et Radiobiologie de la Kératinopoïèse, Institut de Biologie François Jacob, CEA/DRF/IRCM, 91000 Evry, France; socavallero@gmail.com (S.C.); re_neves@yahoo.com.br (R.N.G.); peggy.azzolin@cea.fr (P.A.); 2INSERM U967, 92260 Fontenay-aux-Roses, France; 3Université Paris-Saclay, 75013 Paris 11, France; 4Université Paris-Diderot, 78140 Paris 7, France; 5Ecole Pratique des Hautes Etudes, PSL Research University, UMRS 951, Genethon, 91002 Evry, France; stockho@genethon.fr

**Keywords:** human epidermis, keratinocytes, stem cells, precursor cells, low-dose γ irradiation, regeneration, dysplasia, epithelial-to-mesenchymal transition (EMT), ZEB1

## Abstract

For the general population, medical diagnosis is a major cause of exposure to low genotoxic stress, as various imaging techniques deliver low doses of ionizing radiation. Our study investigated the consequences of low genotoxic stress on a keratinocyte precursor fraction that includes stem and progenitor cells, which are at risk for carcinoma development. Human skin organoids were bioengineered according to a clinically-relevant model, exposed to a single 50 mGy dose of γ rays, and then xeno-transplanted in nude mice to follow full epidermis generation in an in vivo context. Twenty days post-xenografting, mature skin grafts were sampled and analyzed by semi-quantitative immuno-histochemical methods. Pre-transplantation exposure to 50 mGy of immature human skin organoids did not compromise engraftment, but half of xenografts generated from irradiated precursors exhibited areas displaying focal dysplasia, originating from the basal layer of the epidermis. Characteristics of epithelial-to-mesenchymal transition (EMT) were documented in these dysplastic areas, including loss of basal cell polarity and cohesiveness, epithelial marker decreases, ectopic expression of the mesenchymal marker α-SMA and expression of the EMT promoter ZEB1. Taken together, these data show that a very low level of radiative stress in regenerating keratinocyte stem and precursor cells can induce a micro-environment that may constitute a favorable context for long-term carcinogenesis.

## 1. Introduction

The development of human three-dimensional (3D) organoids by bioengineering now permits these models to make an increasing contribution to deciphering developmental processes, tissue and organ physiology and pathophysiological contexts. Notably, achievements have been reported with heart organoids in the domain of myocardial infarction and drug cardiotoxicity [1], with brain organoids for studies of hypoxic brain injury and prematurity [2] and medulloblastoma modeling [3], with liver organoids for studies of normal [4] and cancer [5] development, and with organoids modeling normal and cancer contexts in the digestive tract [6,7]. In skin, human 3D organoids have demonstrated efficiency in the modeling of pathophysiological contexts, such as defects in the epidermal barrier associated with atopic dermatitis [8], or epidermal cancer proneness in xeroderma pigmentosum [9]. Notably, bioengineered 3D epidermises have contributed to the knowledge of keratinocyte stem and progenitor cells [10,11,12,13].

Today, deciphering the adverse impacts of normal tissue exposure to low radiation doses constitutes a biomedical research field of growing interest, due to their increasing use in medical diagnosis technologies such as computed tomography (CT) scans, for both adult and pediatric patients [14,15]. Radiotherapy (RT) is also a source of low-dose exposure for normal tissues and organs surrounding the targeted tumor. As the number of cancer survivors and their lifespans increase thanks to constant improvement of diagnostic methods and medical management, the problem of RT complications is becoming a medical issue of growing importance. Skin is of particular concern regarding RT adverse reactions, as this organ can develop different types of short- and long-term radio-pathologies [16]. Our group and I. Turesson’s group in Sweden have shown that human skin, and notably the epidermis, can develop different types of complications after exposure to high [17] or low [18,19] radiation doses, complications such as erythema, epidermitis, dysplasia, as well as acanthosis and carcinoma in the long-term. However, the contributions of specific target cell populations in these pathophysiological processes still require in-depth studies.

Here, we have investigated the effects of low radiation doses on the capacity of keratinocyte stem and progenitor cells to ensure epidermis regeneration, and have explored the cellular perturbations at the origin of radio-induced disorders that can affect this tissue. We show that pre-transplantation exposure of keratinocyte precursor cells to a single dose of 50 mGy, at the initial stage of 3D epidermis generation, induced focal dysplasia in xenografted epidermises, exhibiting characteristics of epithelial-to-mesenchymal transition (EMT).

## 2. Materials and Methods

### 2.1. Human Tissue and Cell Materials

The present study was approved by the review board of the iRCM (Institut de Radiobiologie Cellulaire et Moléculaire, CEA (French Alternative Energies and Atomic Energy Commission), Fontenay-aux-Roses, France), and is in accordance with the scientific, ethical, safety and publication policy of CEA (CODECO number DC-2008-228, reviewed by the ethical research committee IDF-3). Human skin tissue from adult healthy donors was collected in the context of breast reduction surgery, after informed consent. Epidermal keratinocytes and dermal fibroblasts were extracted after enzymatic treatment. Frozen banked samples of human epidermal holoclone keratinocytes, generated and characterized in [11], were studied as a model of immature skin keratinocyte precursor cells.

### 2.2. Bi-Dimensional Culture of Keratinocytes

Holoclone keratinocyte samples were thawed and amplified in bi-dimensional mass conditions one week before use for skin substitute bioengineering. Cultures were performed in a serum-containing medium, in the presence of a feeder layer of human dermal fibroblasts growth-arrested by γ irradiation (60 Gy), as described in [11]. Plastic devices coated with type I collagen were used (Biocoat, BD Biosciences, Le Pont de Claix, France). Composition of the serum-containing medium included DMEM and Ham’s F12 media (Gibco, ThermoFisher, Les Ulis, France) (*v/v*, 3/1 mixture), 10% fetal calf serum (Hyclone, Fisher Scientific, Illkirch, France), 10 ng/mL epidermal growth factor (EGF) (Chemicon, Fisher Scientific, Illkirch, France), 5 μg/mL transferrin (Sigma, Saint-Quentin Fallavier, France), 5 μg/mL insulin (Sigma, Saint-Quentin Fallavier, France), 0.4 μg/mL hydrocortisone (Sigma, Saint-Quentin Fallavier, France), 180 μM adenine (Sigma, Saint-Quentin Fallavier, France), 2 mM tri-iodothyronine (Sigma, Saint-Quentin Fallavier, France), 2 mM L glutamine (Gibco, ThermoFisher, Les Ulis, France) and 100 U/mL penicillin/streptomycin (Gibco, ThermoFisher, Les Ulis, France).

### 2.3. Three-Dimensional Skin Substitute Bioengineering

Plasma-based human skin substitutes were reconstructed according to [12,20,21]. Human plasma (generous gift from Pr Lataillade, Biomedical Research Institute of French Armies (IRBA), INSERM U1197 Clamart, France), was mixed on ice with 4.68 mg/mL sodium chloride (Fresenius Medical Care, Savigny, France), 0.8 mg/mL CaCl_2_ (Laboratoire Renaudin, Itxassou, France), 9.7 μg/mL Exacyl (tranexamic acid) (Sanofi, France) and human dermal fibroblasts. The mixture was spread in 9.6 cm^2^ Petri dishes (BD Biosciences, Le Pont de Claix, France) and plasma fibrin was allowed to polymerize for 30 min at 37 °C. Fibrin gels were then covered with keratinocyte growth medium (same composition as that used for 2D cultures). The next day, keratinocytes were seeded onto these dermal substrates, at the density of 2400 cells/cm^2^. After 2 weeks of culture (medium changed every 2 days), skin substitutes were ready for xenografting.

### 2.4. Skin Substitute Irradiation

Exposition of skin substitutes to ionizing radiation was performed at the initial step of epidermis regeneration by keratinocyte precursor cells. Accordingly, samples were irradiated 6 h after keratinocyte seeding onto dermal reconstructs, a time sufficient for their recovery from trypsinization and attachment to their new environment. A 137Cs source was used (γ rays, IBL637 irradiator, Cis-Bio international, Saclay, France). Low dose irradiations (50 mGy) were performed at the dose rate of 50 mGy/min, and irradiations at higher dose (2 Gy) at the dose rate of 850 mGy/min. Control samples were sham-irradiated (same processing except γ ray delivery). The control, 50 mGy and 2 Gy cohorts comprise respectively 14, 14 and 13 skin substitutes.

### 2.5. Skin Substitute Xenografting

Experimental procedures [12,21] were approved by the ethical committee CEEA-51 from the Center for Exploration and Experimental Functional Research (CERFE) (Genople^®^, Evry, France). Experiments and housing were managed at CERFE under appropriate aseptic conditions. Immuno-deficient athymic Nude *Foxn1^nu^* mice (ENVIGO, Gannat, France) were used as recipients for the xenografting of human skin substitutes. Mice were anesthetized via intraperitoneal injection of ketamine (Centravet, Maisons-Alfort, France) and xylasine (Centravet, Maisons-Alfort, France), and maintained onto a heated surface to avoid hypothermia. A full-thickness disk of dorsal skin (~1 cm^2^) was removed. This mouse skin piece was then devitalized by serial freezing in liquid nitrogen and thawing, and kept for use as a bio-bandage. The wound bed was covered with an equivalent surface of human bioengineered skin substitute. The devitalized piece of mouse skin was then sutured to the mouse skin border, to cover and transiently protect the xenograft site. This bio-bandage was removed 1 week later under isoflurane (Axience, Pantin, France) anesthesia (anesthetic unit from Minerve, Esternay, France). Mice were euthanized 20 days post-xenografting for analysis of human regenerated skin characteristics using the cervical dislocation method, under anesthesia. Notably, previous characterization of the present xenograft model by live imaging performed on grafts generated with [GFP^+^] transduced keratinocytes has shown no diffuse mixing between the human and mouse epidermises, indicating the absence of recruitment of mouse epithelial cells within human xenografts (Appendix A and [12]).

### 2.6. Preparation of Skin Sections for Histological Characterization

After dissection from euthanized recipient mice, human regenerated skin samples were washed in PBS, and then fixed for 1 day in a buffered solution of 10% formalin (Labonord, Villeneuve D’ascq, France). Fixed tissue samples were dehydrated by graded successive ethanol treatment and then paraffin-embedded. Paraffin sections of 5 µm thickness were prepared (Novaxia histology laboratory, Saint Laurent Nouan, France). For histological characterization, sections were colored with hematoxylin-eosin-saffron (HES), and then scanned and converted into high resolution digital slides using the Axio Scan.Z1 system (Zeiss, Marly le Roi, France) at the imagery platform of the Genethon institute (Evry, France), or using the Pannoramic scan II system (3D Histech, Budapest, Hungary) at the histology platform of INRA-CEA (Jouy-en-Josas, France).

### 2.7. Quantitative Histology

#### 2.7.1. Percentage of Section Length Displaying Abnormal Epidermis Histology

For the determination of section length corresponding to defective regeneration, 6 whole-length HES-stained sections were systematically considered for each individual xenograft. Epidermis areas displaying abnormal organization were identified by the experimenter and their length was measured using a digital scale. Percentages reported on whole section lengths were calculated. Data were expressed as dot plots cumulating the analysis of multiple regions for each section to ensure representativity (numbers are indicated in figure legends).

#### 2.7.2. Detection of Non-Cohesive Spaces within Regenerated Epidermises

For quantification of abnormally regenerated epidermis areas, semi-quantitative estimation of non-cohesive spaces was performed on 6 whole-length HES-stained sections for each xenograft. Digital images were converted into binary pictures in greyscale using Fiji software. After establishing a threshold based on tissue-free areas of slides, expanded intercellular spaces present within regenerated epidermises were converted into red pixels, which were automatically quantified using an in-house routine (imagery platform of the Genethon institute, Evry, France). Dot plots cumulated the analyses of multiple regions for each section (numbers are indicated in figure legends).

#### 2.7.3. Assessment of Keratinocyte Polarity

For each individual xenograft, analysis was performed on 2 sections stained with DAPI (Fluoroshield™, Sigma-Aldrich, Saint-Quentin Fallavier, France), in which 3 regions of interest (ROIs) were defined, corresponding to ~ 800 µm section length. For the analysis focused on areas displaying abnormal organization, these regions were identified by the experimenter and selected as ROIs. A mask was defined by the experimenter to extract the basal keratinocyte layer and characterize nuclei orientation versus the dermo-epidermal junction (JDE) plane. Angle measurements were performed automatically using a routine developed with Fiji software, and data were plotted into 18 angle categories (from 0° to 90°) using R software (Genethon imagery platform, Evry, France).

### 2.8. Section Processing for Immunofluorescence Analyses

Paraffin-embedded sections (Novaxia histology laboratory, Saint Laurent Nouan, France) were deparaffinized in xylene and rehydrated in ethanol-H_2_O. Antigen retrieval was then performed by immersion of paraffin sections for 20 min in Target Retrieval Solution (Dako, Glostrup, Denmark) at 95 °C. Non-specific antibody binding was blocked either by incubation in a 2% BSA (bovine serum albumin) solution or in serum. Staining was performed using non-conjugated primary antibodies, revealed using fluorochrome-conjugated secondary antibodies. Negative controls were performed, corresponding to the staining procedure without primary antibody, and showed no signal. Antibodies and blocking condition are listed in Table 1. Nuclei were stained with DAPI (Fluoroshield™, Sigma-Aldrich, Saint-Quentin Fallavier, France). Image acquisition was performed using a Leica SP8 fluorescence imaging system. For fluorescence semi-quantitative analysis, stained sections were converted into high resolution digital slides using the Axio Scan.Z1 (Zeiss, Marly le Roi, France) at the Genethon imagery platform (Evry, France).

### 2.9. Semi-Quantitative Immunofluorescence Analyses

All marker analyses were performed on 2 stained sections for each xenograft. For analysis of keratinocyte polarity, 3 ROIs were defined per section, corresponding to ~ 800 µm length. Areas displaying abnormal organization were identified by the experimenter and considered as ROIs for their specific characterization. For analysis of VANGL2 expression, a mask was defined to extract the basal keratinocyte layer. VANGL2 signal level (arbitrary units, a.u.) was determined using a routine developed with Fiji software (Genethon imagery platform, Evry, France). DAPI staining of nuclei was used for signal normalization. For analysis of ZEB1 expression, positive keratinocytes were counted by the experimenter, and percentage of ZEB1^+^ cells was calculated. DAPI staining (Fluoroshield™, Sigma-Aldrich, Saint-Quentin Fallavier, France) was used to identify and count all nuclei within epidermises. For analysis of E-cadherin and α-smooth muscle actin (α-SMA) expression, fluorescence levels (a.u.) were determined within epidermises using a routine developed with Fiji software (Genethon imagery platform, Evry, France). Keratin-14 (K14) and β-catenin were used to specifically mark keratinocytes.

### 2.10. Apoptosis Assay

Search for genomic DNA fragmentation associated with late apoptosis was performed in sections of skin substitutes pre-xenografting and regenerated skin 20 days post-xenograting, using the TUNEL (terminal deoxynucleotidyl transferase dUTP nick-end labeling) principle. The In Situ Cell Death Detection Kit was used (Roche Molecular Biochemicals, Mannheim, Germany), according to the manufacturer’s instructions. Technical positive controls corresponded to sections treated for 10 min with 1500 U/mL recombinant DNase I (Roche Molecular Biochemicals, Mannheim, Germany). Nuclei were colored with DAPI (Fluoroshield™, Sigma-Aldrich, Saint-Quentin Fallavier, France). Image acquisition was performed using a Leica SP8 fluorescence imaging system (Leica microsystems, Nanterre, France).

### 2.11. Statistics

Statistical analyses were achieved using GraphPad Prism software (GraphPad Software, San Diego, CA, USA). Statistical significance of the data was determined using the Mann–Whitney U test.

## 3. Results

### 3.1. Experimental Design

A functional approach was designed to model the impact of ionizing radiation (IR) on the regenerative capacity of human epidermal keratinocyte precursor cells (Figure 1). The cellular material used in this study is defined as holoclone keratinocytes (Figure 1A), which correspond to the progeny of single keratinocyte stem cells [11]. These cells exhibit extensive long-term growth potential in bidimensional (2D) culture, as well as genomic stability and efficient epidermal regeneration, as assessed by in vitro epidermis reconstruction and in vivo xenografting [11,12], thus showing functional characteristics of a highly immature population of cultured precursors. Three-dimensional (3D) skin substitutes were bioengineered (Figure 1B) according to a preclinical process [20]. Skin reconstructs were either exposed or not exposed to ionizing radiation (IR) (Figure 1C) at an early stage of epidermis development, corresponding to a non-stratified keratinocyte basal monolayer (see Figure 1B). Samples were submitted to a single exposure to IR at a low dose (50 mGy) or at a higher dose (2 Gy). The next day, irradiated and non-irradiated tissue samples were xenografted onto recipient nude mice (Figure 1D) [21]. This experimental process enabled the study of human epidermis regeneration in an in vivo context up to complete differentiation. Fully differentiated human epidermis substitutes were then characterized (Figure 1E). The model repeatedly gave rise to normally organized and differentiated epidermises.

### 3.2. Irradiation Did Not Compromise Epidermis Regeneration but Induced Local Dysplasia

Wound re-epithelialization was macroscopically observed in all xenografted mice, indicating that keratinocyte precursors were globally functional in the three experimental conditions. Moreover, in xenografted skin sections colored with hematoxylin-eosin-saffron (HES), epidermis stratification and the presence of a horny layer could be observed in all conditions. Quantitative histological analysis was then performed on HES-colored sections to characterize epidermal regeneration (Figure 2). In xenografts performed with non-irradiated keratinocyte precursors (controls), epidermal development was similar to that of a native epidermis in a majority of the observed areas, exhibiting a regular basal layer and correctly stratified supra-basal layers. Notably, differentiated granular and horny layers were clearly identifiable (Figure 2A). In some areas, a discrete disorganization affecting the basal and spinous layers was observed (Figure 2B), which represented an average of 3% of the section length, and corresponded to the background rate of epidermal irregularity of the experimental model. A maximum of 12% irregular areas was observed in two out of 14 control xenografts (Figure 2B). Accordingly, the 12% value was considered a threshold for categorizing abnormal epidermis regeneration. Epidermises regenerated by irradiated keratinocyte precursor cells exhibited local marked disorganization of the basal and spinous layers, alteration of the dermo-epidermis junction and infiltration of keratinocytes in the dermis, which characterized abnormalities termed dysplastic areas (DAs) (Figure 2A). In xenografts performed with keratinocyte precursors that received the IR dose of 2 Gy, DAs represented an average of 10% of the section length, with 10 out of 13 xenografts above the control cohort threshold value, and a maximum DA extent of 42% (*p* < 0.0001 versus non-irradiated) (Figure 2B,C). A quite unexpected observation was the marked impact of the low-dose IR of 50 mGy on regenerated epidermis characteristics—an average of 12% of the section length corresponded to DAs in this condition. Notably, seven out of 14 xenografts were above the control cohort threshold value, with a maximum observed DA extent of 93% (*p* < 0.0001 versus non-irradiated) (Figure 2B,C). Taken together, these observations showed that a single exposure to IR can perturb epidermis regeneration by human keratinocyte precursors, even at a dose of 50 mGy. Importantly, a search for DNA fragmentation (TUNEL assay), performed either on 3D skin substitutes, in vitro 24 h post-irradiation or in vivo 20 days post-xenografting, did not detect any positive signal (Figure 2D,E), documenting an absence of apoptosis. In addition, the presence or absence of p16^INK4a^, which exerts the function of stress-induced senescence promotion in keratinocytes [22], was assessed by immunofluorescence. No p16^INK4a^ staining was observed in keratinocytes, either in normal epidermis areas or in dysplastic regions of xenografts (Appendix A and Appendix A), suggesting that senescence is not a major mechanism in dysplasia and EMT development. The next parts of the study were then focused on samples in the low-dose conditions.

### 3.3. Dysplastic Areas Exhibited Impaired Polarity of Basal Keratinocytes

As an abnormal organization of the keratinocyte basal layer was systematically observed in dysplastic areas, a particular focus was made on this epidermal compartment. In healthy skin, basal keratinocytes are oriented perpendicularly to the dermo-epidermal junction (JDE), whereas loss of polarity occurs in various pathophysiological processes including epithelial-to-mesenchymal transition (EMT). Measurements of basal nuclei orientation were performed on the xenograft sections by image analysis of stained nuclei (Figure 3). Nuclei orientations versus the JDE plane were determined (Figure 3A) and classified according to three categories: nearly perpendicular (angles between 60° and 90°), nearly parallel (angles between 0° and 30°) and oblique (angles between 30° and 90°) (Figure 3B). In control xenografts, a majority of nuclei had a nearly perpendicular orientation, and similar data were obtained in normal areas of xenografts from irradiated keratinocyte precursors. In contrast, a marked increase of oblique and nearly horizontal nuclei orientations was detected within DAs, demonstrating a significant loss of epithelial polarity (*p* < 0.0001 versus repartition in normal areas) (Figure 3B). To further document this observation, the expression pattern of VANGL2, a membrane protein involved in the regulation of cell polarity and migration [23], was then analyzed in xenograft sections by immunofluorescence (Figure 3C,D). This protein was expressed in all basal keratinocytes in normal epidermis, whereas it was reduced or absent in dysplastic cells (Figure 3C). Semi-quantitative image analysis confirmed that the VANGL2 level was significantly reduced in DAs (*p* < 0.0001 versus control areas) (Figure 3D). In summary, a loss of basal keratinocyte polarity that spatially correlated with a perturbated expression of VANGL2 was identified as a marked characteristic of dysplastic areas.

### 3.4. Defective Cell–Cell Interactions Were Observed in Dysplastic Areas

In a non-pathological context, the interfollicular epidermis forms a cohesive structure devoid of large intercellular spaces. Microscopic observation of xenograft sections pointed out the presence of visible spaces within DAs, the extent of which were then estimated by semi-quantitative image analysis using an in-house algorithm (Figure 4A,B). Analysis of the whole section length showed that empty spaces were globally augmented in xenografts generated with irradiated keratinocyte precursors (*p* < 0.005 versus control xenografts) (Figure 4B). When the analysis was focused on DAs, this parameter was even more significantly increased, due to the presence of large non-cohesive zones (*p* < 0.0001 versus control xenografts and versus normal areas from the 50 mGy conditions) (Figure 4A,B). Considering the regulatory link between VANGL2 and E-cadherin [24], the expression patterns of the latter cell–cell adhesion molecule were analyzed in xenograft sections by immunofluorescence (Figure 4C,D). In xenografts generated with non-irradiated keratinocyte precursors (Figure 4C), as well as outside DAs in xenografts from irradiated cells, a typical expression pattern of E-cadherin on keratinocyte membranes was observed, thinly contouring cells within the basal and supra-basal layers. In DAs, a more blurred pattern, associated with a globally lower expression level, was observed (Figure 4C). Semi-quantitative analysis of fluorescence signals confirmed the marked reduction of E-cadherin level in DAs (*p* < 0.0001 versus control xenografts; *p* < 0.01 versus whole-length 50 mGy xenograft sections) (Figure 4D). In summary, these observations identified defective connectivity as a characteristic of dysplastic areas.

### 3.5. Epithelial-to-Mesenchymal Transition Markers Were Detected in Epidermal Dysplastic Areas

Considering the major importance of E-cadherin and stable cellular adherens junctions in the pathophysiological process of epithelial-to-mesenchymal transition (EMT) [25], expression of EMT effectors and markers was analyzed in xenografts. Firstly, expression of the transcription factor ZEB1 was investigated (Figure 5A,B). In control xenografts (Figure 5A), as well as in normal regions of xenografts from irradiated keratinocyte precursors (not shown), ZEB1 expression was almost exclusively restricted to the dermis, and rarely observed in epidermal keratinocytes (Figure 5A). Quantification performed on whole-length sections indicated that ZEB1-positive cells represented about 3% or less of most sections (Figure 5B). In contrast, observation of xenografts from irradiated keratinocyte precursors showed that ZEB1-positive keratinocytes were abundantly present in DAs (Figure 5A), reaching up to 18% of keratinocytes (*p* < 0.0001 versus control xenografts, and versus whole-length 50 mGy xenograft sections) (Figure 5B). Of note, this increase was not significant when considering whole-length 50 mGy xenograft sections, compared to controls (Figure 5B), suggesting that activation of ZEB1 expression did not concern all keratinocytes. Of note, β-catenin expression on keratinocyte membranes was also impaired in DAs. Expression of α-smooth muscle actin (α-SMA), a typical mesenchymal marker, was then analyzed in association with keratin-14 (K14), used as a specific counterstaining of basal keratinocytes (Figure 5C,D). In control xenografts (Figure 5C), as well as in normal regions of xenografts from irradiated keratinocyte precursors (not shown), α-SMA was exclusively distributed within the dermis. This distribution was strongly modified in DAs, with the detection of α-SMA signaling in cells that co-expressed K14 (Figure 5C), showing an ectopic expression in keratinocytes. α-SMA signal quantification focused on DAs showed a detectable signal above control values (*p* < 0.0001 versus control xenografts, and versus whole-length 50 mGy xenograft sections) (Figure 5D). As for ZEB1, the level of α-SMA did not appear to be significantly affected when considering whole-length 50 Gy xenograft sections, compared to the background signal of controls (Figure 5D). Taken together, these results identified a significant link between the pathophysiological process of EMT and characteristics of dysplastic areas. 

## 4. Discussion

High radiation doses such as those delivered during radiation therapy (RT) produce pathological changes in mesenchymal tissues with long-term alterations of fibroblast phenotypic and functional characteristics that may impair the quality of life of treated cancer patients [16,26,27], whereas the pathophysiology of epithelial complications of RT has been investigated less [17]. Furthermore, the effects of very low doses of genotoxic stress are poorly documented [19,28], although dysplasia has been reported in the skin of radiotherapy patients [18]. We have here addressed the question of possible adverse reactions subsequent to the exposition of keratinocyte stem and progenitor cells to low radiation doses. The dose analyzed here (50 mGy) is in the range of those delivered to normal tissues adjacent to the target tumor volume during radiotherapy, and is relevant for biomedical diagnostic procedures, notably scanner imaging. It is much lower than the dose limit accepted for the induction of carcinoma, which has been proposed to be around 500 mGy, notably based on the Japanese atomic bomb survivor study [29]. The key contribution of the present study was the demonstration that dysplasia and epithelial-to-mesenchymal transition (EMT) develop in epidermises generated by keratinocyte stem and precursor cells exposed to a single low dose of γ irradiation, thus documenting a micro-environment favoring the development of skin cancer.

Epidermal holoclone keratinocytes provided a model representative of an immature population of cultured precursors containing functional stem and progenitor cells. These cells correspond to the clonal progeny of single keratinocyte stem cells that were functionally characterized according to an extensive growth potential exceeding 100 population doublings through successive subcultures, and the capacity for epidermis reconstruction in vitro and regeneration in vivo [11,12]. Importantly, the stem cell status attributed to holoclones has been demonstrated in vivo by cellular tracing in the entirely regenerated epidermis of an epidermolysis bullosa patient engrafted with an autologous, genetically corrected skin substitute [30]. Moreover, the fibrin-based epidermis organoids that were used in this study corresponded to an adaption from a clinically relevant model of bioengineered skin substitute [20,31]. The present epidermis regeneration approach permits modeling of skin stem and progenitor cell properties and potentialities in conditions characterized by a higher level of cell proliferation and metabolic activities than in the context of healthy skin homeostasis. These conditions probably exacerbate radiosensitivity and thus allow observation of cell and tissue responses to low stress levels.

The semi-quantitative imaging approaches that were set-up to characterize the epidermal distribution patterns of molecular markers provided clues for the understanding of genotoxic stress-induced epidermal dysplasia and EMT development. Firstly, the marked alteration of VANGL2 patterns that was detected in association with the impaired orientation of basal keratinocyte nuclei constituted a relevant parameter, due to the involvement of this membrane protein in the regulation of cell polarity and migration [23]. VANGL2 is a central component of the planar cell polarity signaling pathway (PCP), which is essential for correct epidermal development and morphogenesis [32,33], as well as epidermal wound repair [34]. The loss of epithelial polarity is a key process in the early steps of EMT. Weakening and disruption of cell–cell contacts, which were documented here by the presence of non-cohesive spaces and a local decrease in E-cadherin level, are typical characteristics of the pathophysiological process of EMT [25]. Finally, detection of an ectopic expression of the mesenchymal marker α-SMA in keratinocytes consolidated the EMT-like phenotypic switch occurring in radiation-induced dysplastic areas (DAs) [35]. Among the various transcription factors involved in EMT, ZEB1 has been described as a major early player in its development, later favoring epithelial tumor progression in association with E-cadherin suppression [36]. We show here the appearance of ectopic ZEB1 protein in keratinocytes of dysplastic epidermis, thus providing a link between low-dose irradiation of holoclone keratinocyte precursor cells and potential initiation sites of the carcinoma development cascade. Notably, perturbated functions of the ‘wingless’ (WNT) signaling pathways, as suggested here by the loss of the β-catenin protein, associated with focal dysplasia, might constitute a promoter event of the pathophysiological processes described here.

In conclusion, we have developed an approach based on in vitro bioengineering of human skin organoids, coupled with in vivo xenografting in immune-deficient mice, to explore the pathophysiological consequences of low-dose γ-irradiation exposure of epidermal stem and progenitor cells on their subsequent regenerative capacity. We have observed that a single 50 mGy radiation dose was sufficient to promote local perturbations in regenerated epidermises with cellular and molecular characteristics of dysplasia and EMT, which may constitute an initial risk for the future development of carcinomas. Interestingly, this approach is directly applicable to other biomedical research domains, for example characterization of the skin’s defenses and responses to pollution [37].

## Figures and Tables

**Figure 1 cells-09-01912-f001:**
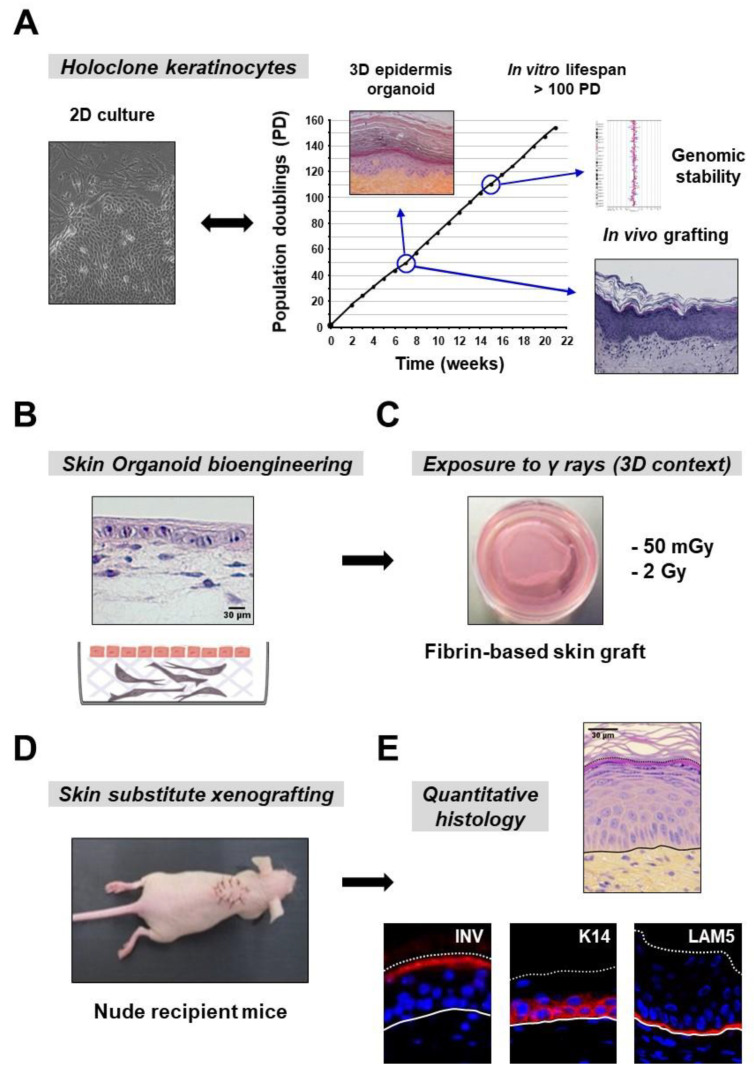
Study architecture. (**A**) Holoclone keratinocytes were used as a cellular model of cultured human epidermal precursor cells. (**B**) Bioengineering of an immature three-dimensional (3D) human skin substitute using holoclone keratinocytes for epidermis regeneration. A typical section colored with hematoxylin-eosin-saffron (HES) is shown. (**C**) Single-exposure of 3D human skin substitutes to ionizing radiation (IR): 50 mGy (dose rate: 50 mGy/min), 2 Gy (dose rate: 850 mGy/min) or sham irradiation. (**D**) The next day, xenografting of irradiated and non-irradiated 3D human skin substitutes in recipient nude mice, which enables full maturation of human epidermises in an in vivo context. (**E**) Removal and sampling of human grafts 20 days post-xenografting for quantitative histology and analysis of marker expression patterns. Pictures from a typical section with normal histology (HES coloration), and normal expression pattern of epidermal markers are shown: keratin-14 (K14), laminin-5 (LAM5) and involucrin (INV).

**Figure 2 cells-09-01912-f002:**
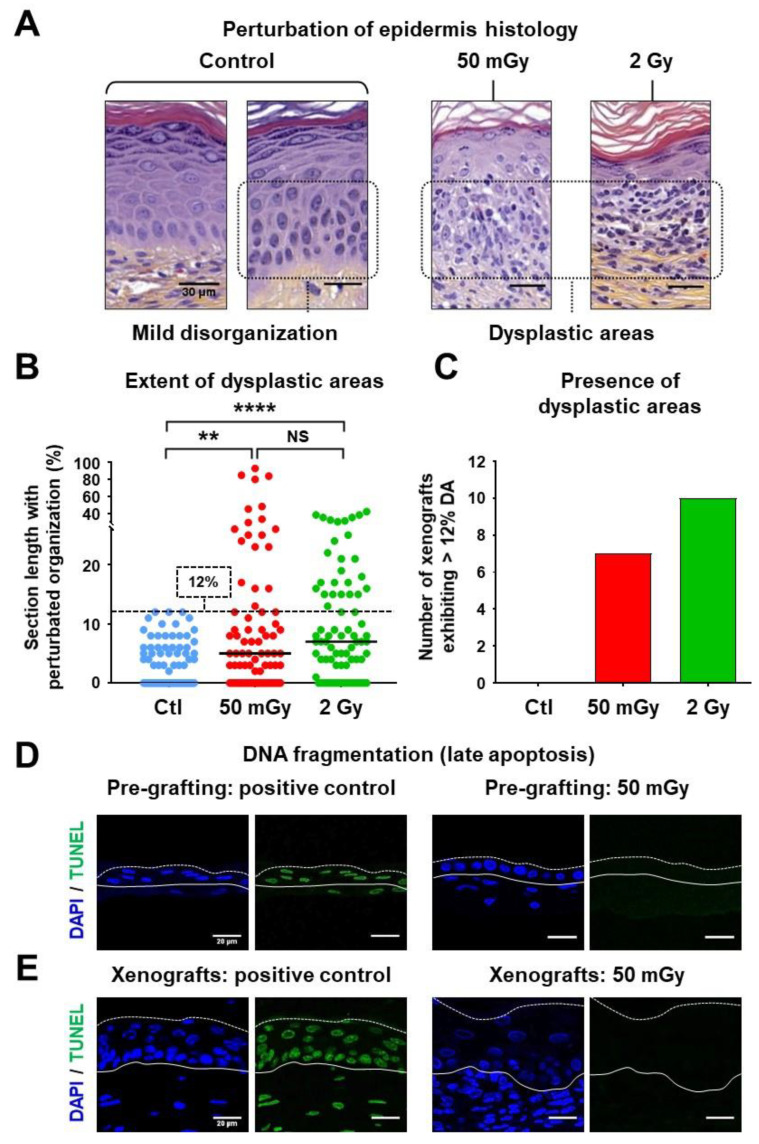
Local dysplastic areas developed in xenografted epidermises originating from irradiated keratinocyte precursors. (**A**) HES coloration of human skin samples 20 days post-xenografting. Representative pictures were selected for the visualization of the normal histology of control epidermises, the mild disorganization considered as the background of the xenograft model, and examples of dysplastic areas (DA) that were characteristic of irradiated conditions. (**B**) Estimation of the percentage of tissue section length displaying mild disorganization or DA. A total of 14 xenografts were performed for the control (Ctl) and 50 mGy conditions, and 13 were performed for the 2 Gy condition. Dot plots cumulated the analyses of 6 sections for each xenograft. Bars correspond to median values (NS *p* > 0.05; ** *p* < 0.01; **** *p* < 0.0001, Mann–Whitney U test). (**C**) The histogram shows the numbers of xenografts that displayed DAs corresponding to at least 12% of epidermis section length: *n* = 7 out of 14 xenografts for the 50 mGy condition; *n* = 10 out of 13 xenografts for the 2 Gy condition. (**D**,**E**) Search for genomic DNA fragmentation associated with late apoptosis using the terminal deoxynucleotidyl transferase dUTP nick-end labeling (TUNEL) assay in pre-grafting skin substitutes (24 h post-irradiation) (**D**) and 20 days post-xenografting (**E**). Technical positive controls corresponded to sections treated with DNase. Nuclei were colored with DAPI. No signal was detected either in the sham-irradiated or in the irradiated conditions at both experimental stages. Photographs shown are representative of *n* = 14 xenografts for both the control and 50 mGy conditions.

**Figure 3 cells-09-01912-f003:**
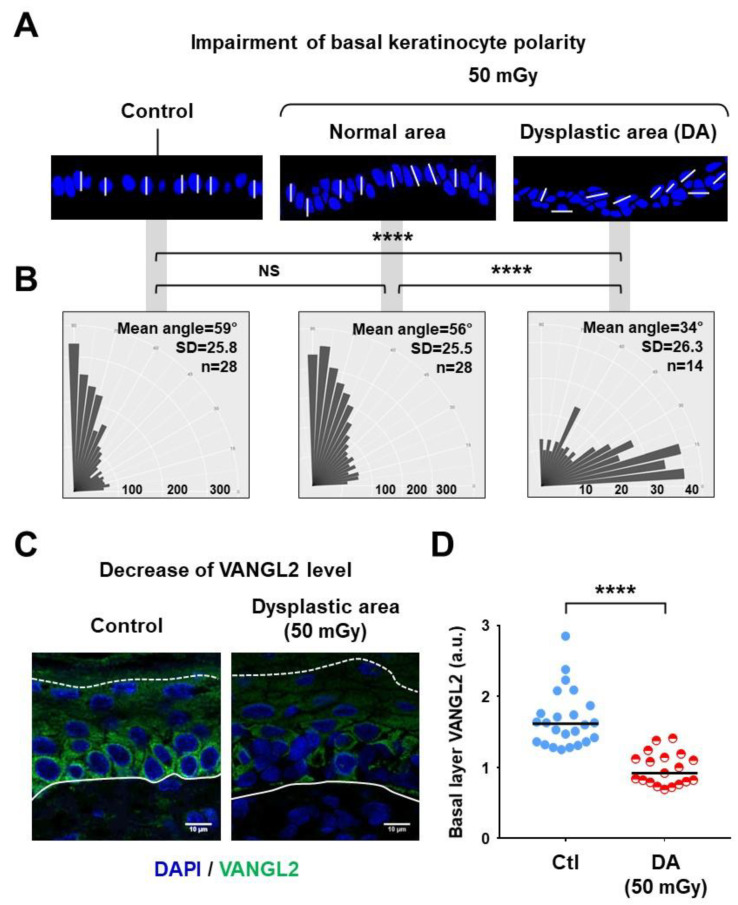
Perturbations of basal keratinocyte polarity in dysplastic epidermis areas of xenografts. (**A**) Imaging of basal keratinocyte nuclei orientation versus the dermo-epidermal junction (JDE) plane, after coloration with DAPI. Typical zoomed pictures of basal layer sections are shown, with white bars added to illustrate some perpendicular, oblique and parallel nuclei orientations. (**B**) Distribution of basal keratinocyte nuclei according to angle versus the JDE plan into 18 angle categories from 0° to 90°, characterized by automated image analysis. The vertical axis represents angle values and the horizontal axis numbers of cells in the different angle categories. Analysis was performed on 14 different xenografts for all conditions, *n* indicates numbers of analyzed dysplastic areas (NS *p* > 0.05; **** *p* < 0.0001, Mann–Whitney U test). (**C**) Immunofluorescence detection of VANGL2 protein. Representative pictures were selected for the visualization of VANGL2 in basal keratinocytes from normal epidermis regions, showing its decrease or absence in basal keratinocytes from dysplastic areas. Nuclei were colored with DAPI. (**D**) Quantification of VANGL2 fluorescence in the epidermis basal layer (arbitrary units, a.u.), showing a reduced signal within DAs. Dot plots cumulated the analyses of 24 normal regions from 14 control (Ctl) xenografts and 19 DAs from the 14 xenografts of the 50 mGy conditions. Bars correspond to median values. (**** *p* < 0.0001, Mann–Whitney U test).

**Figure 4 cells-09-01912-f004:**
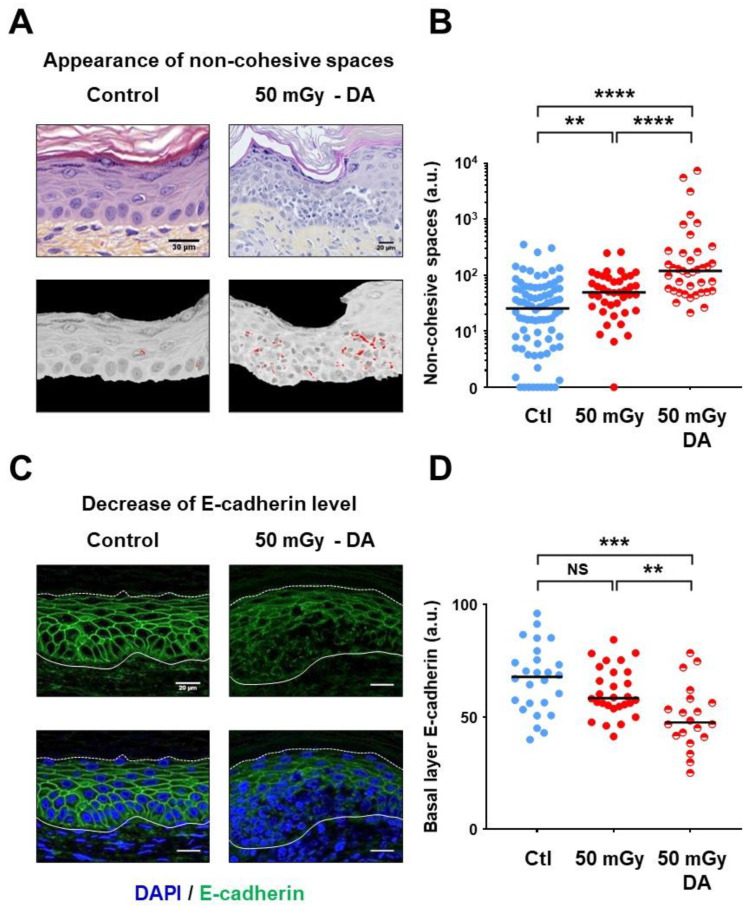
Defective cell-cell cohesiveness in dysplastic areas. (**A**) Typical pictures of HES sections illustrating cohesive epidermis in normal regions and the presence of visible non-cohesive spaces within DAs (top panel). Conversion of spaces into red pixels by automated image processing (bottom panel) for semi-quantitative analysis. (**B**) Semi-quantitative analysis of non-cohesive spaces based on red pixel conversion (arbitrary units, a.u.), revealing the presence of significant non-cohesive zones in DAs. Dot plots cumulated the analyses of 82 normal regions from control (Ctl) xenografts, 41 random regions from the 50 mGy xenografts and 39 selected regions corresponding to DAs in the 50 mGy xenografts. Bars correspond to median values (** *p* < 0.01; **** *p* < 0.0001, Mann–Whitney U test). (**C**) Immunofluorescence detection of E-cadherin. Representative pictures were selected for the visualization of E-cadherin expression in normal epidermis regions, showing its decrease or absence in basal keratinocytes from DAs. Nuclei were colored with DAPI. (**D**) Semi-quantitative analysis of E-cadherin (arbitrary units, a.u.), showing a lower signal within DAs. Dot plots cumulated the analyses of 26 normal regions from the 14 control (Ctl) xenografts and 20 DAs from the 14 xenografts of the 50 mGy condition. Bars correspond to median values (NS *p* > 0.05; ** *p* < 0.01; **** *p* < 0.0001, Mann–Whitney U test).

**Figure 5 cells-09-01912-f005:**
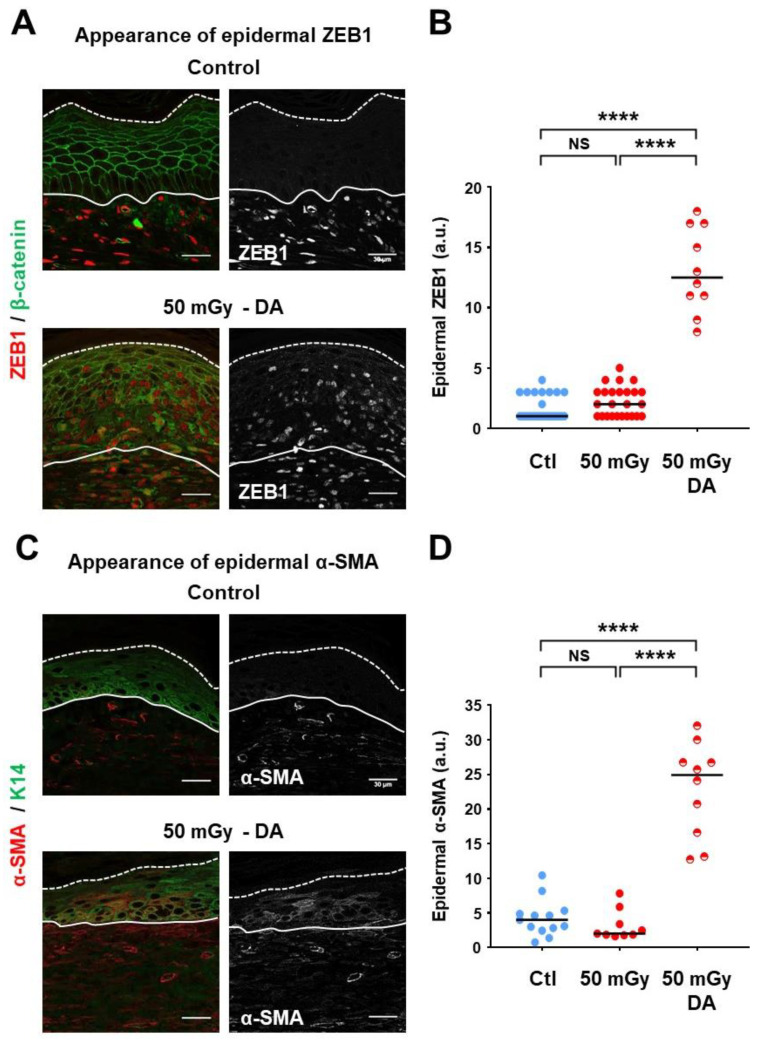
Detection of epithelial-to-mesenchymal transition markers in dysplastic areas. (**A**) Immunofluorescence detection of ZEB1. Representative pictures were selected for the visualization of dermal ZEB1 expression in normal skin regions and its ectopic presence in epidermal DAs. Gray-tone visualization of ZEB1 signal is also shown. (**B**) Quantitative analysis showed rare ZEB1-positive keratinocytes in normal epidermis regions, and abundant ZEB1-positive keratinocytes in DAs. Dot plots cumulated the analyses of 26 normal regions from the 14 control (Ctl) xenografts, 25 random regions from the 14 xenografts of the 50 mGy condition and 10 selected regions corresponding to DAs in 50 mGy xenografts. Bars correspond to median values (NS *p* > 0.05; **** *p* < 0.0001). β-catenin staining, which marked keratinocyte contours, revealed impaired epidermal organization and its reduced expression. (**C**) Immunofluorescence detection of α-smooth muscle actin (α-SMA). Representative pictures were selected for the visualization of dermal α-SMA expression in normal skin regions and its abnormal presence in epidermal DAs. Sections were stained for keratin 14 to mark basal keratinocytes. Gray-tone visualization of α-SMA signal is also shown. (**D**) Dot plots cumulated the analyses of 13 normal regions from control (Ctl) xenografts, 9 random regions from the 50 mGy xenografts and 10 selected regions corresponding to DAs in 50 mGy xenografts. Bars correspond to median values (NS *p* > 0.05; **** *p* < 0.0001).

**Table 1 cells-09-01912-t001:** Antibodies.

Primary Antibodies	Blocking Reagents
Rabbit polyclonal anti-ZEB1 [H-102] (sc-25388, Santa Cruz, Heidelberg, Germany)	BSA
Rabbit polyclonal anti-involucrin (ab53112, Abcam, Paris, France)	BSA
Rabbit polyclonal anti-αSMA (ab5694, Abcam, Paris, France)	Serum
Mouse monoclonal anti-lamin 5 (ab78286, Abcam, Paris, France)	Diagomics, Blagnac, France
Mouse monoclonal anti-cytokeratin 14 [LL002] (Leica Biosystems, Nanterre, France)	BSA
Mouse monoclonal anti-βcatenin [15B8] (ab6301, Abcam, Paris, France)	Serum
Mouse monoclonal anti-E-cadherin [M168] (ab76055, Abcam, Paris, France)	Serum
**Secondary Antibodies**	
Goat anti-Mouse, Alexa Fluor^®^594 conjugate (A-11032, ThermoFisher scientific, Les Ulis, France)	
Goat anti-Mouse, Alexa Fluor^®^488 conjugate (A-11001, ThermoFisher scientific, Les Ulis, France)	
Goat anti-Rabbit, Alexa Fluor^®^594 conjugate (A-11037, ThermoFisher scientific, Les Ulis, France)	
Donkey anti-Rabbit, FITC conjugate (ab97063, Abcam, Paris, France)	

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
