# Peer review of "Exposure of Human Skin Organoids to Low Genotoxic Stress Can Promote Epithelial-to-Mesenchymal Transition in Regenerating Keratinocyte Precursor Cells"

_cells, 2020, doi:10.3390/cells9081912_

Round 1

Reviewer 1 Report

The manuscript entitled "Exposure of human skin organoids to low genetic stress can promote epithelial-to-mesenchymal transition in regenerating keratinocyte precursor cells" is an experimental papers that discussed generated organoids with the derm cells obtained from the patients, which were xenografted into nude mice. Following the exposure of the cells to low dose gamma radiation, the transplanted graft was shown to change their focus from epithelial lineage to mesenchymal lineage. The manuscript is well-written with sufficient described methods and rationale, there are a couple of minor issues

  1. Please do not mention the overall result of the study at the end of the introduction (except if it is the journal format). Please wrap the introduction with the goal of the study.
  2. Some of the methods are repeated/mentioned in the results section again, please provide all the details related to the methodology of the study in the methods section.
  3. Please be consistent with "DAPI" vs dapi
  4. Liter is usually mention in upper case as L (except if the journal format required to be l).

Overall this is a good study with clear methods and discussions pertaining to the results obtained.

The above mentioned minor changes needs to be addressed prior to publication.

Author Response

cells-887539 - Responses to Reviewer 1

cells-887539 - Responses to Reviewer 1

Black bold font: reviewer’s comments.

Black font: author’s responses.

Red font: changes in the main manuscript text.

Reviewer 1.

The manuscript entitled "Exposure of human skin organoids to low genetic stress can promote epithelial-to-mesenchymal transition in regenerating keratinocyte precursor cells" is an experimental papers that discussed generated organoids with the derm cells obtained from the patients, which were xenografted into nude mice. Following the exposure of the cells to low dose gamma radiation, the transplanted graft was shown to change their focus from epithelial lineage to mesenchymal lineage. The manuscript is well-written with sufficient described methods and rationale, there are a couple of minor issues.

Overall, this is a good study with clear methods and discussions pertaining to the results obtained.

We thank Reviewer 1 for his positive appreciation of our work.

Please do not mention the overall result of the study at the end of the introduction (except if it is the journal format). Please wrap the introduction with the goal of the study.

According to this request, the following sentences have been removed from the ‘Introduction’ end:

Human holoclone keratinocytes, corresponding to an immature population of cultured epidermal precursor cells (i.e. stem and progenitor cells), were used for the in vitro reconstruction of 3D skin organoids, which were then xenografted in recipient nude mice to enable the follow-up of full regeneration of human epidermis in an in vivo context.

Some of the methods are repeated/mentioned in the results section again, please provide all the details related to the methodology of the study in the methods section.

According to this request, we have revised the paragraph 3.1. of ‘Results’ to remove the information that was redundant with ‘Materials and Methods’:

A functional approach was designed to model the impact of ionizing radiation (IR) on the regenerative capacity of human epidermal keratinocyte precursor cells (Figure 1). The cellular material used in this study is defined as holoclone keratinocytes (step A), which correspond to the progeny of single keratinocyte stem cells that were isolated from healthy skin biopsies and used to initiate clonal cultures [11]. These cells exhibit extensive long-term growth potential in bidimensional (2D) culture, genomic stability, and efficient epidermal regeneration, as assessed by in vitro epidermis reconstruction and in vivo xenografting [11,12], thus showing functional characteristics of a highly immature population of cultured precursors. Plasma-based Three-dimensional (3D) skin substitutes were bioengineered (step B) using holoclone keratinocytes as precursor cells for epidermis regeneration, and human skin fibroblasts embedded into a plasma gel as a dermal reconstruct, according to a preclinical process [20]. Skin reconstructs were exposed or not to ionizing radiation (IR) (step C) at an early stage of epidermis development, corresponding to a non-stratified keratinocyte basal monolayer (see B). Samples were submitted to a single exposure to IR (γ rays, cesium source) at a low dose (50 mGy) or at a higher dose (2 Gy), or were sham-irradiated (controls). The next day, irradiated and non‑irradiated tissue samples were xenografted onto a wound bed created by full-thickness skin excision in the dorsal region of recipient nude mice (step D) [21]. This experimental process enabled the study of human epidermis regeneration in an in vivo context up to complete differentiation. Fully differentiated human epidermis substitutes were removed and sampled 20 days post-xenografting, and then characterized according histological and cellular parameters, which was combined to the analysis of molecular marker expression patterns (step E). The model repeatedly gave rise to normally organized and differentiated epidermises.

Please be consistent with "DAPI" vs dapi.

"dapi" been replaced by "DAPI" in paragraph 2.7. of ‘Materials and Methods’.

Liter is usually mention in upper case as L (except if the journal format required to be l).

"l" has been replaced by "L" (Materials and Methods)

The above-mentioned minor changes need to be addressed prior to publication.

We hope the revisions that have been made will meet reviewer’s expectation.

Reviewer 2 Report

In the current manuscript, authors elegantly investigated the effects of low dose irradiation on engineered skin xenografts. They showed that low dose irradiation negative affected xenografts with focal dysplasia and EMT phenotype, which might contribute to long-term carcinogenesis. It is very interesting study.  However, there are few shortcomings to be addressed as below:

  1. In Figure 2D and E, authors showed that no DNA fragments were observed in pre-grafting and xenografts after 50 mGy exposure. These data indicate that other factors mediated the epithelial-to-mesenchymal transition (EMT) in xenografts. It will be plus if authors can examine the status of cellular apoptosis and senescence in xenografts, which will provide some potential mechanisms for EMT.
  2. In Figure 3, authors measured expression of VANGL2 in xenografts showing that polarity in dysplastic areas was disrupted. It would be better to include another marker to confirm the data, such as CDC42, etc.  
  3. Currently, authors checked day 20 time point in xenografts. It will be interesting to see whether dysplasia areas have carcinogenesis in longer time point.

Author Response

cells-887539 - Responses to Reviewer 2

Black bold font: reviewer’s comments.

Black font: author’s responses.

Red font: changes in the main manuscript text.

Reviewer 2.

In the current manuscript, authors elegantly investigated the effects of low dose irradiation on engineered skin xenografts. They showed that low dose irradiation negative affected xenografts with focal dysplasia and EMT phenotype, which might contribute to long-term carcinogenesis. It is very interesting study.

We thank Reviewer 2 for his positive appreciation of our work.

However, there are few shortcomings to be addressed as below.

We hope the revisions that have been made will meet reviewer’s expectation.

In Figure 2D and E, authors showed that no DNA fragments were observed in pre-grafting and xenografts after 50 mGy exposure. These data indicate that other factors mediated the epithelial-to-mesenchymal transition (EMT) in xenografts. It will be plus if authors can examine the status of cellular apoptosis and senescence in xenografts, which will provide some potential mechanisms for EMT.

To take this comment into consideration, we have assessed the presence or absence of p16 INK4a in xenografts by immunofluorescence, as its expression is associated with stress-induced keratinocyte senescence. Three tissue contexts were analyzed: non-irradiated control, normal areas from the 50 mGy condition, and dysplastic areas from the 50 mGy condition (timing). No p16 INK4a staining was observed in keratinocytes, either in normal epidermis areas or in dysplastic areas. These new data are presented as ‘Supplementary Materials’ (Figure S2 and Table S1). They support the proposition that stress-induced senescence was not a major mechanism in dysplasia and EMT development.

The following text has been added in paragraph 3.2. of ‘Results’:

In addition, the presence or absence of p16 INK4a, which exerts the function of stress-induced senescence promoter in keratinocytes [22], was assessed by immunofluorescence. No p16 INK4a staining was observed in keratinocytes, either in normal epidermis areas or in dysplastic regions of xenografts (Figure S2 and Table S1), suggesting that senescence is not a major mechanism in dysplasia and EMT development.

A bibliographical reference referring to p16 INK4a has been added: Sasaki et al., ref [22].

In Figure 3, authors measured expression of VANGL2 in xenografts showing that polarity in dysplastic areas was disrupted. It would be better to include another marker to confirm the data, such as CDC42, etc. 

Our belief is that a major observation is the disruption of basal keratinocyte nuclei orientation versus the dermo-epidermal junction plane, as shown in Figure 3A and 3B, but we agree that deciphering the molecular mechanisms responsible for this perturbation of tissue organization constitutes by itself a very relevant topic. Characterizing this aspect would require in-depth investigations that would justify a specific study.

Currently, authors checked day 20 time point in xenografts. It will be interesting to see whether dysplasia areas have carcinogenesis in longer time point.

The xenograft system used in this study was primarily designed for the modeling of the early stages of human epidermis in vivo regeneration, and is not optimized in its present form for long-term follow-up of grafted epidermis evolution. Further developments and adaptation of the system will be necessary to study the evolution of dysplastic areas in longer kinetics, in the perspective of modeling carcinogenesis.

Reviewer 3 Report

In this study, Cavallero et al analyze the effect of radiation in skin regeneration from epidermal progenitor cultures. They find that radiation leads to the formation of dysplastic areas with elevated markers of EMT in transplanted skin equivalents. Although the study is interesting, it seems preliminary and does not provide a mechanistic understanding of the underlaying processes.

  • Although the premise of studying keratinocyte behavior following irradiation is interesting, the study design seems to be measuring the effect of radiation in epidermal regeneration. According to the authors, samples were irradiated 6 hours after keratinocyte seeding which measures the effect of radiation in a monolayer. It is not clear what this would correspond in the normal skin setting that is alluded in the introduction, were patients are receiving low doses of radiation in full skin. Why was this point selected? Why the radiation was not performed in full grown skin equivalents? How does this translate into a human setting were patients are receiving radiation to normal, full thickness skin? Why was the transplantation into mice needed? Skin equivalents can be grown in vitro in a liquid-air interface.
  • The authors correlate the dysplasia observed following regeneration of irradiated cultures as a “micro-environment favoring the development of skin cancer”. Again, since this is an artificial system, there is not good correlation between this system and what would happen in a human setting. Are dysplastic regions observed following low dose radiation in normal human skin? Experiments are performed in immunocompromised mice. It is probable that dysplastic keratinocytes would be detected by the immune system and eliminated in normal settings. The conclusions of the authors seem overstated.
  • No control is performed to show the composition of human fibroblasts and keratinocytes in the xenografts. Human specific antigens should be used to confirm that the area analyzed is composed of human cells. Some of the antibodies used are targeted to human, but due to the high conservation of the proteins analyzed between human and mouse, control staining showing positive areas only in the human xenograft but not in the sourronding mouse skin should be included.
  • More evidence showing the mechanism by which radiation leads to the activation of EMT and dysplasia is needed.
  • Even though no apoptosis is observed, radiation can lead to the activation of senescent pathways that could explain some of the observed phenotypes. Analysis of senescent markers and potential role of senescence in the formation of the dysplastic areas would improve the quality of the study.
  • Co-staining with basal keratinocyte markers like keratin 5 or TP63 is needed to confirm that the changes observed are related to keratinocyte progenitor cells, particularly in the experiments of cell polarity, VANGL2 and alpha-SMA expression. The dysplastic areas show accumulation of very small cells and disruption of basal membrane, are these cells keratinocytes?

Author Response

cells-887539 - Responses to Reviewer 3

Black bold font: reviewer’s comments.

Black font: author’s responses.

Red font: changes in the main manuscript text.

Reviewer 3.

In this study, Cavallero et al analyze the effect of radiation in skin regeneration from epidermal progenitor cultures. They find that radiation leads to the formation of dysplastic areas with elevated markers of EMT in transplanted skin equivalents. Although the study is interesting, it seems preliminary and does not provide a mechanistic understanding of the underlaying processes.

Although the premise of studying keratinocyte behavior following irradiation is interesting, the study design seems to be measuring the effect of radiation in epidermal regeneration. According to the authors, samples were irradiated 6 hours after keratinocyte seeding which measures the effect of radiation in a monolayer. It is not clear what this would correspond in the normal skin setting that is alluded in the introduction, were patients are receiving low doses of radiation in full skin. Why was this point selected? Why the radiation was not performed in full grown skin equivalents? How does this translate into a human setting were patients are receiving radiation to normal, full thickness skin? Why was the transplantation into mice needed? Skin equivalents can be grown in vitro in a liquid-air interface.

The authors correlate the dysplasia observed following regeneration of irradiated cultures as a “micro-environment favoring the development of skin cancer”. Again, since this is an artificial system, there is not good correlation between this system and what would happen in a human setting. Are dysplastic regions observed following low dose radiation in normal human skin? Experiments are performed in immunocompromised mice. It is probable that dysplastic keratinocytes would be detected by the immune system and eliminated in normal settings. The conclusions of the authors seem overstated.

It is difficult to investigate effects of low dose irradiation in normal human skin, for evident ethical reasons, and it is also not permitted to investigate directly on radiotherapy patient skin, for medical reasons. However, I Turesson obtained the authorizations in Sweden, and she set up a bank of skin samples removed from patients during a radiotherapy time-course. Interestingly, this author obtained samples from areas far from the tumor, thus receiving low doses of IR (0.1, 0.2, and 0.45 Gy). She demonstrated that repeated low doses exert deleterious effects on skin, and described hypersensitivity of keratinocytes in these conditions. Notably, the 2010 article shows in figure 1 typical images of epidermal dysplasia after 0.08 and 0.47 Gy, see reference below. Although obtained in different conditions, the data on patients support our findings. The aim of our study was different from Turesson’s, as we aimed to investigate the effects of a single low dose exposure on keratinocyte stem and progenitor cells of normal skin, and notably on their intrinsic capacity of skin regeneration. As no established marker can identify keratinocyte stem cells in vivo, the cell populations of interest must be firstly isolated to allow investigation, and thus skin grafting was the most appropriate model.

However, we agree with reviewer that we cannot directly extrapolate to in vivo conditions and that conclusions cannot be too much overstated. This is why we used a careful conclusion: “These data show that a very low level of radiative stress in regenerating keratinocyte stem and precursor cells can induce a micro-environment that may constitute a favoring context for long-term carcinogenesis.”

A sentence has been added to the discussion, pointing on dysplasia in patient skin, and we thank the reviewer for helping in manuscript improvement.

Furthermore, the effects of very low doses of genotoxic stress are poorly documented [19,28], although dysplasia has been reported in the skin of radiotherapy patients [18].

A low-dose hypersensitive keratinocyte loss in response to fractionated radiotherapy is associated with growth arrest and apoptosis. Turesson I, Nyman J, Qvarnström F, Simonsson M, Book M, Hermansson I, Sigurdardottir S, Johansson KA. Radiother Oncol. 2010 Jan;94(1):90-101. doi: 10.1016/j.radonc.2009.10.007. Epub 2009 Nov 20. PMID: 19931928

No control is performed to show the composition of human fibroblasts and keratinocytes in the xenografts. Human specific antigens should be used to confirm that the area analyzed is composed of human cells. Some of the antibodies used are targeted to human, but due to the high conservation of the proteins analyzed between human and mouse, control staining showing positive areas only in the human xenograft but not in the sourronding mouse skin should be included.

To characterize the xenograft model, xenograft site coverage was assessed by live imaging, performed on grafts generated with [GFP+] transduced human keratinocytes using a fluorescence confocal laser endomicroscopy technology. No diffuse mixing between the human and mouse epidermises was observed, indicating the absence of recruitment of mouse epithelial cells within human xenografts.

A supplemental Figure has been added (Figure S1), associated with corresponding methodological details (Supplementary Methods).

The following text has been added in paragraph 2.4. of ‘Materials and Methods’ (main manuscript):

Notably, previous characterization of the present xenograft model by live imaging performed on grafts generated with [GFP+] transduced keratinocytes has shown no diffuse mixing between the human and mouse epidermises, indicating the absence of recruitment of mouse epithelial cells within human xenografts (Figure S1 and [12]).

More evidence showing the mechanism by which radiation leads to the activation of EMT and dysplasia is needed. Even though no apoptosis is observed, radiation can lead to the activation of senescent pathways that could explain some of the observed phenotypes. Analysis of senescent markers and potential role of senescence in the formation of the dysplastic areas would improve the quality of the study.

To take this comment into consideration, we have assessed the presence or absence of p16 INK4a in xenografts by immunofluorescence, as its expression is associated with stress-induced keratinocyte senescence. Three tissue contexts were analyzed: non-irradiated control, normal areas from the 50 mGy condition, and dysplastic areas from the 50 mGy condition (timing). No p16 INK4a staining was observed in keratinocytes, either in normal epidermis areas or in dysplastic areas. These new data are presented as ‘Supplementary Materials’ (Figure S2 and Table S1). They support the proposition that stress-induced senescence was not a major mechanism in dysplasia and EMT development.

The following text has been added in paragraph 3.2. of ‘Results’:

In addition, the presence or absence of p16 INK4a, which exerts the function of stress-induced senescence promoter in keratinocytes [22], was assessed by immunofluorescence. No p16 INK4a staining was observed in keratinocytes, either in normal epidermis areas or in dysplastic regions of xenografts (Figure S2 and Table S1), suggesting that senescence is not a major mechanism in dysplasia and EMT development.

A bibliographical reference referring to p16 INK4a has been added: Sasaki et al., ref [22].

Co-staining with basal keratinocyte markers like keratin 5 or TP63 is needed to confirm that the changes observed are related to keratinocyte progenitor cells, particularly in the experiments of cell polarity, VANGL2 and alpha-SMA expression. The dysplastic areas show accumulation of very small cells and disruption of basal membrane, are these cells keratinocytes?

The epithelial nature of epidermal cells in dysplastic areas (DA) was documented by the detection of two keratinocyte markers: keratin 14 (K14) and β-catenin (Figure 5A and 5C). We agree that in-depth phenotypic and molecular characterization of DA-keratinocytes constitutes a next important step arising from the present study.

Round 2

Reviewer 3 Report

The manuscript is improved by the inclusion of new data. Responses to review are acceptable.